# Hybrids of Sterically Hindered Phenols and Diaryl Ureas: Synthesis, Switch from Antioxidant Activity to ROS Generation and Induction of Apoptosis

**DOI:** 10.3390/ijms241612637

**Published:** 2023-08-10

**Authors:** Elmira Gibadullina, Margarita Neganova, Yulia Aleksandrova, Hoang Bao Tran Nguyen, Alexandra Voloshina, Mikhail Khrizanforov, Thi Thu Nguyen, Ekaterina Vinyukova, Konstantin Volcho, Dmitry Tsypyshev, Anna Lyubina, Syumbelya Amerhanova, Anna Strelnik, Julia Voronina, Daut Islamov, Rakhmetulla Zhapparbergenov, Nurbol Appazov, Beauty Chabuka, Kimberley Christopher, Alexander Burilov, Nariman Salakhutdinov, Oleg Sinyashin, Igor Alabugin

**Affiliations:** 1Arbuzov Institute of Organic and Physical Chemistry, FRC Kazan Scientific Center, Russian Academy of Sciences, Akad. Arbuzov St. 8, Kazan 420088, Russia; neganova83@mail.ru (M.N.); yulia.aleks.97@mail.ru (Y.A.); microbi@iopc.ru (A.V.); khrizanforov@iopc.ru (M.K.); aplyubina@gmail.com (A.L.); syumbelya07@mail.ru (S.A.); nikanna@iopc.ru (A.S.); burilov_2004@mail.ru (A.B.); oleg@iopc.ru (O.S.); ialabugin@gmail.com (I.A.); 2Institute of Physiologically Active Compounds at Federal Research Center of Problems of Chemical Physics and Medicinal Chemistry, Russian Academy of Sciences, Severnij Pr. 1, Chernogolovka 142432, Russia; yandulovacaterina@gmail.com; 3The Department of General Organic and Petrochemical Synthesis Technology, The Kazan National Research Technological University, Karl Marx St. 68, Kazan 420015, Russia; nhbtran1912@gmail.com (H.B.T.N.); nguyenthithu202@mail.ru (T.T.N.); 4Department of Medicinal Chemistry, Novosibirsk Institute of Organic Chemistry, Lavrentiev Av. 9, Novosibirsk 630090, Russiatsypyshev@nioch.nsc.ru (D.T.); anvar@nioch.nsc.ru (N.S.); 5Kurnakov Institute of General and Inorganic Chemistry of the Russian Academy of Sciences, Leninskii Prospekt, 31, Moscow 119071, Russia; juliavoronina@mail.ru; 6Laboratory for Structural Analysis of Biomacromolecules, Kazan Scientific Center of Russian Academy of Science, 31, Kremlevskaya, Kazan 420008, Russia; daut1989@mail.ru; 7Laboratory of Engineering Profile, Department of Engineering Technology, Korkyt Ata Kyzylorda University, 29A, Aiteke Bi Street, Kyzylorda 120014, Kazakhstan; nurasar.82@mail.ru; 8Department of Chemistry and Biochemistry, Florida State University, 95 Chieftan Way, Tallahassee, FL 32306-3290, USA; bchabuka@fsu.edu (B.C.);

**Keywords:** sterically hindered phenol, quinone methides, urea, anticancer activity, cytotoxicity, mitochondrial membrane potential, apoptosis, glycolysis, molecular docking, electrochemical oxidation

## Abstract

The utility of sterically hindered phenols (SHPs) in drug design is based on their chameleonic ability to switch from an antioxidant that can protect healthy tissues to highly cytotoxic species that can target tumor cells. This work explores the biological activity of a family of 45 new hybrid molecules that combine SHPs equipped with an activating phosphonate moiety at the benzylic position with additional urea/thiourea fragments. The target compounds were synthesized by reaction of iso(thio)cyanates with C-arylphosphorylated phenols containing pendant 2,6-diaminopyridine and 1,3-diaminobenzene moieties. The SHP/urea hybrids display cytotoxic activity against a number of tumor lines. Mechanistic studies confirm the paradoxical nature of these substances which combine pronounced antioxidant properties in radical trapping assays with increased reactive oxygen species generation in tumor cells. Moreover, the most cytotoxic compounds inhibited the process of glycolysis in SH-SY5Y cells and caused pronounced dissipation of the mitochondrial membrane of isolated rat liver mitochondria. Molecular docking of the most active compounds identified the activator allosteric center of pyruvate kinase M2 as one of the possible targets. For the most promising compounds, **11b** and **17b**, this combination of properties results in the ability to induce apoptosis in HuTu 80 cells along the intrinsic mitochondrial pathway. Cyclic voltammetry studies reveal complex redox behavior which can be simplified by addition of a large excess of acid that can protect some of the oxidizable groups by protonations. Interestingly, the re-reduction behavior of the oxidized species shows considerable variations, indicating different degrees of reversibility. Such reversibility (or quasi-reversibility) suggests that the shift of the phenol-quinone equilibrium toward the original phenol at the lower pH may be associated with lower cytotoxicity.

## 1. Introduction

The challenge in the development of antitumor drugs is to achieve the selective death of tumor cells with no toxic effects on the normal tissues. A promising approach for achieving this goal and creating drugs with selective antineoplastic action is based on using specific conditions that exist in tumors. For example, malignant neoplasms can have a more acidic pH than normal cells. The acidification occurs because the transformed cells rely on glycolysis as a key energy pathway. In this pathway, the conversion of glucose to lactic acid causes acidification of the environment. This phenomenon, known as the Warburg effect, is thought to be one of the fundamental metabolic changes in tumors [1,2,3]. Therefore, molecules containing motifs capable of pH-dependent transformation can be used for selective targeting of neoplastic cells [4,5,6]. Furthermore, the efficiency of such tunable antitumor drugs may be increased by direct modulation of the glycolysis process.

Another characteristic feature of tumors is a high concentration of reactive oxygen species, which cause and propagate genetic abnormalities [7]. Phenolic compounds are capable of modulating the process of oxidative stress and switching the type of activity [8,9,10]. This property accounts for the increasing role of phenols as important structural components in the construction of smart drugs. Sterically hindered phenols (SHPs) represent a class of known phenolic antioxidants that have compounds with strong anticancer activity [11,12,13,14,15] as well as activity against other diseases associated with oxidative stress [16]. Such compounds have a dual “chameleonic” character. Under normal conditions, they effectively protect cellular membranes from the damaging action of reactive oxygen species (ROS) and can reduce the toxic effects of drug therapy. However, at high ROS levels, phenolics are converted into reactive quinone methides [17,18,19], which have a destructive effect on lipids, proteins and DNA, resulting in the death of tumor cells. This duality of SHPs [10,20] seems ideal for the development of new drugs but, in order to harness the high reactivity of quinone methides and avoid undesired effects on the healthy cells, one needs to achieve spatiotemporal control of their localization and activation.

To construct new anticancer compounds in the presented work, we pursued a hybrid approach that combined the following conceptual elements presented in Figure 1:(a)Regulation of the efficiency of quinone methide generation in vivo by introducing electron-withdrawing phosphoryl groups at the benzylic position of SHPs;(b)Modification of compounds with diaryl urea fragments known to interfere with glycolysis pathways, the important energy source for cancer metabolism.

Quinones are redox-active molecules that can undergo multiple redox cycles mediated by their anion radicals, which leads to the formation of ROS (Figure 1) [21,22,23]. The reactivity of *p*-alkyl-substituted phenols can be regulated by introducing various substituents at the benzylic carbon, which significantly affect the rate of initial hydrogen atom transfer. Although the formation of the phenoxy radical is believed to be the rate-determining stage, the presence of an EWG phosphonate fragment on the methylene group assists in the formation of quinone methides and their anion radicals by further activating the doubly benzylic C-H bond. It is well known that the phosphonate group can increase the acidity of C-H by more than 13 orders of magnitude [24,25]. Additionally, the presence of phosphonate moieties has been used to improve the bioavailability of known anticancer drugs [26,27,28]. Furthermore, adding a Ph group at the benzylic carbon of toluene increases the C-H acidity by 9 pKa units (Figure 1D). Together, the effects of all three substituents should greatly acidify the benzylic C-H bond. Intriguingly, the calculated O-H BDFE (Bond Dissociation Free Energies) in p-cresol decrease from 78 to 39 kcal/mol upon C-H deprotonation (Figure 1E), explaining how C-H acidity can be coupled to the antioxidant activity of such phenols. The important added benefit of C-H deprotonation is that it also paves the way for the formation of quinone methide radical anions, a reactive intermediate that can transfer an electron to molecular oxygen with the simultaneous formation of two reactive species (the quinone methide and the superoxide anion).

Functional groups capable of selective binding to proteins upconverted in tumor cells can increase the selectivity of tumor targeting. In this context, urea derivatives offer an opportunity to establish a network of drug–target interactions through the formation of hydrogen bonds [29,30,31,32,33]. Such interactions can be an important element of molecular recognition and bioactivity. The NH fragment behaves as a hydrogen bond donor, while the oxygen or the sulfur atom of urea acts as an acceptor (Figure 1). For example, the urea fragment plays a significant role in molecular recognition and inhibition at the active center of intracellular kinases and cell surface receptor tyrosine kinases such as vascular endothelial growth factor receptors, as observed with the anticancer drugs Sorafenib and Lenvatinib. Due to the density of H-bond fragments and synthetic versatility [34,35,36], ureas play a significant role in creating promising small molecule drugs that can selectively bind to tumor cells and, under the influence of various factors, exert the desired effect.

Thus, our aim is to develop antitumor compounds which have cytotoxic effects on tumor cells but exhibit only antioxidant activity in healthy cells. As a platform for combining these components, we used diarylmethylphosphonates containing SHP fragments, which we had obtained in high yields by addition of activated electron-rich aromatics at the electrophilic carbon of α-phosphorylated quinone methides [37]. The presence of terminal amino groups in these structures allows for the introduction of various active moieties and opens broad prospects for the development of hybrid antitumor drugs.

## 2. Results

### 2.1. Chemistry

To obtain the target compounds, we synthesized the key intermediates—diarylmethylphosphonates containing SHPs and terminal amino groups. These compounds are powerful electrophiles that can alkylate activated electron-rich aromatics without catalysts. In the synthesis of ureas, we used the classic method based on the interaction of isocyanates with primary amines, as illustrated in Figure 1. Chloroform was chosen as the solvent where urea derivatives precipitate during formation, which simplifies the isolation of the reaction product.

Disubstituted urea derivatives **5**–**17** are formed cleanly when diamines **3a**–**d** and **4a**,**b**,**d** react at room temperature with 8–10 equivalents of aryl isocyanates. Despite using the same large excess of isocyanate, the reactions of 4-chlorophenyl isocyanate with derivatives of 2,6-diaminopyridine **3a** and 4-nitrophenyl isocyanate with **3a**,**b** were incomplete and formed difficult to separate mixtures of mono- and disubstituted products, which we were unable to isolate individually.

We isolated crystals of monosubstituted urea derivatives **7a**,**b** from a DMSO solution of a mixture of mono- and disubstituted reaction products upon evaporation. Their molecular structure obtained from X-ray crystallography is shown in Figure 2. Compounds **7a**,**b** crystallize in the centrosymmetric space group P-1. The bond lengths, valence and torsion angles in the molecules of compounds **7a**,**b** in the crystal are in the ranges of characteristic values for the corresponding types of bonds. It is shown that the molecular structure for the compounds is stabilized by the intramolecular N11H11…N5 H-bond. The crystal packing of these compounds is mediated by a large number of H-bonds (see Appendix A for a more detailed discussion).

Independent of conditions, 2,6-diaminopyridine derivatives **3a**–**d** react with only one mole of phenylisothiocyanate to form monosubstituted thiourea derivatives exclusively and in high yields (**18a**–**d,**
Figure 1). On the other hand, the more reactive diaminobenzenes **4a**,**b**,**d** can form disubstituted derivatives **19a**,**b**,**d** with an excess (e.g., eight equivalents) of phenylisothiocyanate. It should be noted that the reaction time in the thiourea synthesis increased up to 15 h, even when heating the reaction mixture in chloroform.

The ^1^H, ^13^C and ^31^P NMR spectra were fully consistent with the proposed structures. Additionally, the structures of compounds **5a**,**c**,**d**; **6a**,**b**; **18a**,**b**,**d** and **19a**,**b**,**d** were confirmed using 2D correlation NMR experiments (^1^H-^1^H COSY, ^1^H-^13^C HSQC, ^1^H-^13^C HMBC; see Appendix A). The benzylic CH proton is clearly resolved in ^1^H spectra due to its intrinsic chemical shift and is also readily identified by the presence of ^2^J_PH_ constants of ca.25 Hz. ^1^H-^13^C HMBC correlations along aromatic rings and adjacent substituents were established. The principal HMBC correlations of compounds **5a**,**c**,**d**; **6a**,**b**; **18a**,**b**,**d** and **19a**,**b**,**d** (see Appendix A) also show connections between the methyl group at the phosphorus atom and the aromatic ring carbons and between the NH protons of ureas and carbon atoms of the aromatic system. For compound **5d**, a mixture of stereoisomers, likely originating from restricted rotation, is observed. The ^1^H-^31^P HMBC correlation spectrum (Figure 3) of this compound contains a set of cross-peaks characterizing the interaction of the phosphorus atom with protons of the phenoxy group, methyl group, spatially hindered phenol and 2,6-diaminopyridine fragment.

Thus, the interaction of C-arylphosphorylated derivatives of 2,6-diaminopyridine and 1,3-diaminobenzene with arylisocyanates and phenylisothiocyanate opened access to a family of phosphorus-containing SHPs with aryl urea or phenyl thiourea appendages. In the next step, the synthesized compounds were tested for cytotoxicity in vitro against human tumor and normal cell lines.

### 2.2. Biological Evaluation

A wide selection of biological properties was studied for the synthesized compounds in order to determine their antitumor potential and to gain insights into the mechanisms of their action. In particular, we studied the antioxidant properties (both in vitro and in selected assays), the effect on metabolism (glycolysis of tumor cells), the modulating effect on the functional characteristics of isolated rat liver mitochondria and the direct pro-apoptotic effect.

#### 2.2.1. Cytotoxic Profile

To start, the new compounds were tested for cytotoxicity against cancer and normal cell lines. Table 1 presents cytotoxic activity as the IC_50_ values (the concentration of compound that causes 50% cell death in the test population) for the most active compounds. Upon 24 h incubation of cell lines with the tested compounds, a pronounced death of the tumor cells was observed, as evidenced by the IC_50_ values less than 15 μM for compounds **8b**, **14b**, **17b** (against human duodenal adenocarcinoma HuTu 80) and **14b** (against human breast adenocarcinoma MCF-7).

It is important to note that for some compounds there was a decrease in the toxic effect on the non-transformed Chang liver cell line, which can obviously indicate a possible selective effect of the substances, specifically on neoplastic cells. Compounds **5a**,**c**,**d**; **6a**,**b**,**d**; **8a**,**c**,**d**; **9a**,**b**,**c**; **10a**,**c**,**d**; **11c**,**d**; **12c**,**d**; **13d**; **14a**,**d**; **15a**,**d**; **16a**,**d**; **17d**; **18a**,**b**,**d** and **19a**,**b**,**d** are not shown in Table 1 as they had no cytotoxic effect against cells of tumor origin (M-HeLa, MCF-7, HuTu 80, SH-SY5Y) and their IC_50_ exceeded 70 μM.

#### 2.2.2. Analysis of the Antioxidant Potential of Compounds: Switch from Antioxidants to Oxidants

To characterize the antioxidant potential of the active compounds, their effect on the process of lipid peroxidation in rat brain homogenates was studied using a modified variant of the TBA assay in a tablet format. Additionally, to obtain more specific information on the possible mechanism of antioxidant action, we studied the radical-trapping activity of the key compounds in the 2,2-diphenyl-1-picrylhydrazyl (DPPH) and Oxygen Radical Absorbance Capacity (ORAC) assays. Figure 4 shows the data on the antioxidant potential of the synthesized compounds. At 100 μM concentration, the antioxidant activity in the lipid peroxidation test for the most reactive ureas, **17b**, **5b** and **11b**, was similar to that of their parent amines (**3c** and **3d**). All of these compounds inhibited lipid peroxidation from 65% to 99% (Figure 4). This result indicates that antioxidant properties mainly originate from the phenol moiety that is unchanged upon the transformation of free amines into their urea and thiourea derivatives.

However, analysis of the concentration dependency of the antioxidant effect revealed interesting differences between these compounds. In particular, all ureas are better antioxidants than their parent amines at the lower concentrations because they have a much smaller “induction range” of concentrations where no antioxidant activity is observed.

When exploring the possible mechanisms of antioxidant action, it was shown that all the tested compounds have direct radical quenching activity. In the DPPH test, the highest value (64.8 ± 0.7%) was found for **17b**. This activity greatly (by factors of 2 and 4) exceeded the activity of the parent SHP derivatives **3c** and **3d**.

In addition, most of the tested substances displayed levels of oxygen radical absorbance capacity at the level of Trolox, a popular phenolic water-soluble antioxidant. In the case of the DPPH test, compound **17b** was the most active, with a Trolox equivalent value of 1.27.

However, the above results were obtained under nearly neutral conditions (pH 7.4) and simulated oxidative stress using chemical reactions, as well as a model of biological membranes, albeit derived from the brain of laboratory animals. To approximate the conditions in a living organism more closely, we studied the effect of the tested substances on the level of free radicals directly in the tumor cells. Figure 5 shows the results for compounds **11b** and **17b,** which had the most pronounced antioxidant ability. The data clearly indicate that intracellular ROS level in HuTu 80 cells after 48 h of incubation significantly increased upon exposure to **11b** and **17b** at maximum concentration ((670.7 ± 7.2%) and (400.2 ± 5.2%), respectively).

Thus, the analysis of the most effective cytotoxic compounds found high antioxidant activity expressed as the ability to suppress the lipid peroxidation process of rat brain homogenates. The antioxidant activity of cytotoxic compounds was further supported by direct antiradical activity in the ORAC and DPPH tests. However, under conditions of oxidative stress at the cellular level, compound **17b** increases the formation of peroxyl radicals in cells. Like chameleons, the new phenol urea hybrids switch their nature—from antioxidants (outside of cells) to oxidants (inside of tumor cells). This remarkable change reflects the intrinsic redox duality of the phenol/quinone methide redox pair (presented in Figure 1) which qualifies them as “smart biological agents”—protecting healthy tissues under the conditions of low oxygen radical stress and switching to a cell-damaging “warhead” under oxidative stress in the tumors.

#### 2.2.3. Analysis of the Effect of Synthesized Compounds on Tumor Cell Metabolism—Glycolysis and the Activity of the Key Allosteric Enzymes of This Process

It is well known that tumor cells have a proliferative metabolic phenotype in which they exhibit increased glycolytic activity in contrast to the normal cells in a body. This difference indicates the important role of metabolic reprogramming in the progression of oncopathologies and stimulates the search for substances that inhibit the process of glycolysis. Such compounds promise to restore the energy metabolism of transformed cells from the predominant glycolytic pathway to oxidative phosphorylation and restrict the continuous proliferation of tumor cells.

We studied the ability of compounds to inhibit aerobic glycolysis using the Agilent Seahorse XF96e Analyzer (Seahorse Bioscience, North Billerica, MA, USA) assay of cellular metabolism on the level of proton production in samples examined on the SH-SY5Y neuroblastoma cell line using glycolysis stress tests. The extracellular acidification rate of the medium was measured in real time, which allowed us to assess the intensity of glycolysis in the cells by recording the main parameters of glycolytic function: basal glycolysis, glycolytic capacity and glycolytic reserve. Glycolytic capacity describes the maximum amount of glycolysis/glucose breakdown the cells can perform acutely, whereas glycolytic reserve is defined as the difference between the basal and maximal glycolytic capacity.

Glycolysis in SH-SY5Y tumor neuroblastoma culture as the main neoplastic cell energy state in the presence of compounds **11b**, **14b** and **17b** led to a decrease in the rate of extracellular acidification when saturating amounts of glucose were added to the system and thereby suppressed glycolytic function (Figure 6b). When oligomycin was administered, an ATP synthase inhibitor that blocks the last step of oxidative phosphorylation, significant acidification of the medium was observed in the control group, indicating intense glycolysis and lactic acid release. In contrast, for **11b**, **14b** and **17b**, a decrease in glycolytic capacity was observed, which is obviously related to the suppression of lactate production by these substances. Moreover, for two compounds, **11b** and **17b**, the ability to decrease glycolytic reserves was also found.

Thus, the highest efficacy in suppressing the glycolytic function of SH-SY5Y neuroblastoma cell cultures was observed for **11b** and **17b**. For a more detailed understanding of which molecular mechanisms encourage glycolysis inhibition to occur, we performed a molecular docking procedure into the binding sites of pyruvate kinase (PDB ID: 4G1N) and hexokinase (PDB ID: 5HEX). Despite the fact that there are quite a lot of enzymes that catalyze the reactions of the glycolytic pathway, it is hexokinase and pyruvate kinase that are speed-limiting ones [38,39]. Therefore, targeting these enzymes is a promising strategy in the search for modulators of metabolic function in a tumor cell [38,39]. Consequently, our goal was to gain deeper insights into the interaction between the compounds and the allosteric centers of the enzymes.

The data obtained by molecular docking to the binding site of glucose-6-phosphate hexokinase 2 are presented in Table 2. The pose of reference ligand 2-[(3-bromobenzene-1-carbonyl)amino]-6-{[(4-carboxy-5-methylfuran-2-yl)sulfonyl]amino}-2,6-dideoxy-alpha-D-glucopyranose was reproduced with an RMSD of 1.3849.

All compounds studied through molecular docking of the glucose-6-phosphate hexo-kinase 2 binding site reproduced receptor–ligand positions. However, the resulting Docking Score does not meaningfully exceed that of the reference ligand, from which we can conclude that the binding site of glucose-6-phosphate-hexokinase 2 is most likely not the main target of the compounds under study.

The data obtained by molecular docking of the compounds at the activator allosteric site PKM2 are presented in Table 3. The position of the reference ligand N-(4-{[4-(pyrazin-2-yl)piperazin-1-yl]carbonyl}phenyl)quinoline-8-sulfonamide was reproduced with an RMSD of 1.57.

During docking to the activator binding site of pyruvate kinase 2 (PKM2), all compounds reproduced the mode of docking. It should be noted that for compound **17b**, only the R isomer produced a viable docking pose. It is also noteworthy that all compounds had a Docking Score above that of the reference ligand, which may indicate their possible higher binding affinity compared to the comparison drug. Figure 7 shows the possible location of the compounds under study at the center of the allosteric activator pyruvate kinase PKM2.

The calculated binding geometries underscore the ability of urea moieties to participate in multiple H-bonding interactions with the active sites of enzymes. For example, all three substrates in Figure 7 engage in N-H…O=C bonding with the protonated Lys311 moiety at the active site. Furthermore, the aromatic rings of the diaryl urea unit participate in p/p interactions with the aromatic residues of the protein.

The ability of **11b** and **17b** to inhibit glycolysis and to affect the activity of an enzyme catalyzing the key step in the glycolytic pathway is certainly beneficial for their antitumor activity.

#### 2.2.4. Study of the Effect of Compounds on Mitochondrial Characteristics

The capacity to influence the functional characteristics of mitochondria, i.e., the organelle that plays a key role in activating the cascade of cell death through apoptosis, can contribute to the antitumor activity of the compounds. As we know, in the neoplastic process, the key element for uncontrolled proliferation and the maintenance of tumor cell growth is the suppression of apoptotic processes that should protect the organism by killing the transformed cells.

The transmembrane potential of isolated organelles was considered as a parameter of mitochondrial function for the action of potential antitumor agents. The transmembrane potential is a fundamental characteristic that induces selective membrane permeability and controls the initiation of the cell death cascade. Several molecules in this group were found to affect mitochondrial characteristics and, in particular, depolarize the mitochondrial membrane. Compounds **11b** and **17b** were the most active, reaching the 50% level of depolarization at a concentration of 100 μM. Moreover, this activity is concentration-dependent. Figure 8 shows that membrane potential dissipation rate increases with the concentration of tested compounds. Thus, nearly 100% depolarization was observed for **17b** at a concentration of 200 μM.

#### 2.2.5. Pro-Apoptotic Effect of Compounds **11b** and **17b**

As discussed above, the leading SHP/urea hybrids, **11b** and 1**7b**, combine selective toxicity towards tumor cell lines with being effective antioxidants capable of inhibiting glycolysis in transformed SH-SY5Y cells and depolarizing the mitochondrial membrane, thereby exhibiting a potential pro-apoptotic effect. For these substances, the ability to induce apoptosis in HuTu 80 cells was directly investigated.

Mitochondria in eukaryotic cells are major components of respiration and play a key role in protection against oxidative stress-induced damage. As we discussed earlier, maintaining the level of Ψm is important for efficient ROS interception and for cytoprotective properties against aponecrotic events caused by excess ROS. Therefore, the influence of lead compounds on mitochondrial membrane potential and intensification of the apoptosis process were investigated using the fluorescent dye JC-1. JC-1 is a Ψm- sensitive marker that aggregates in the mitochondrial matrix and emits a red fluorescent signal in healthy cells. When Ψm decreases, JC-1 changes to a monomeric state, exhibiting green fluorescence. As expected, 48 h incubation of HuTu 80 cells with the studied substances at concentrations equal to ½ IC_50_ and IC_50_ cytotoxicity revealed a marked decrease in Ψm, which was detected by flow cytometry (Figure 9). In addition, this effect was concentration-dependent (Figure 9).

Thus, these results, along with the earlier experiments on isolated mitochondria, suggest that an additional mechanism of antitumor action in compounds **11b** and **17b** is the ability to induce apoptosis through the internal mitochondrial pathway.

### 2.3. Electrochemical Measurements

Cyclic voltammetry is commonly used to characterize the reduction ability and electrochemical behavior of phenolic antioxidants [40,41]. Generally, there is a correlation between antioxidant and pro-oxidant activity and oxidation potential. However, the connection between the ability to donate an electron and the ability to intercept radicals is not always straightforward and exceptions do exist [42], especially when multiple redox-active groups are present.

Since the SHP/urea hybrids contain several redox-active functionalities, we have used cyclic voltammetry (CV) to understand their mutual influence on each other. At the first stage, we compared the CV of the key SHP subunits to evaluate the influence of each fragment (Figure 10). As the structure becomes more complex, the potential shifts to a more anodic region. The oxidation potential of the parent ionol is known to be in the range of 1.45 V (in CH_3_CN vs. Ag/AgCl, see Figure 10), while diphenyl urea is oxidized at a slightly higher potential (1.60 V vs. Ag/AgCl). Introduction of the electron-withdrawing phosphonate group at the benzylic position very slightly shifts the first oxidation wave without dramatically changing its shape. On the other hand, oxidation of the full SHP/urea hybrid (blue wave in Figure 10) is more complex, suggesting the possibility of several oxidation steps (^ox^E_p_^1^ = 1.56; ^ox^E_p_^2^ = 1.87).

The electrochemical experiments were carried out in acetonitrile solution with the addition of increasing amounts of trifluoroacetic acid. Protonation by the acid can protect several of the functional groups from oxidation, which can simplify analysis of the complex multicomponent voltammograms.

Some of the findings are intriguing. For example, the two SHP/urea hybrids **13b** and **14b** differ only in the position of a Me group at the terminus (i.e., *meta* vs. *para*), though **14b** is around five times more cytotoxic than **13b**. The origin of this difference is, without doubt, complex and may not originate from a single factor. However, intriguingly, the CV experiments suggest that these two compounds are also quite different electrochemically.

Compound **13b** has two characteristic oxidation peaks (see Appendix A), which are particularly visible in the semi-differential form (see Appendix A). When the acid is added, the appearance of the oxidation curves changes significantly. Addition of 30 eq of acid leads to a quasi-reversible wave (see Appendix A), which in turn indicates that under acidic conditions the phenol–quinone equilibrium can be shifted back toward the parent phenol by re-reduction. It is possible that the quick restoration of the phenol prevents **13b** from triggering the mechanisms described in Figure 1.

On the other hand, a structurally similar compound, **14b**, shows a fundamentally changed electrochemical behavior (see Appendix A), which is revealed upon the addition of acid. Since there are as many active redox centers in **13b** as in **14b**, we see sequential oxidation in these compounds with several peaks with similar potentials. Measurements in the presence of the excess of trifluoroacetic acid reveal that the fragment belonging to the OH group does not differ in terms of oxidation potential. However, it differs greatly at the re-reduction part of the curve. Even in the presence of the acid, oxidation remains irreversible (see Appendix A), which is especially apparent in the semi-differential CV.

This observation suggests that it is possible to quickly scan for potential effective substances by using inverse semi-differential linear voltammetry (iSDLSV) and assessing the reversibility of the oxidation process (Figure 11). In Figure 11, the iSDLSV of an effective substance, **14b**, is fundamentally different from iSDLSV for its structural analogue, **13b**, differing by only one substituent at the remote terminus.

Thus, the nature of electrochemical oxidation waves underscores the multifunctional character of the SHP/urea hybrids. Oxidation of the molecule results in several oxidized states originating from the sequential electron transfer steps. Many of these steps are irreversible, highlighting the high reactivity of such species towards oxidants. Protonation can deactivate some of these redox transitions, providing a simpler picture which is helpful for disentangling the individual contributions (see Appendix A for a more detailed discussion).

## 3. Discussion

Currently, there is a significant amount of data indicating that phenolic compounds possess pronounced antioxidant activity. This activity is attributed to their ability to maintain endogenous enzymatic antioxidant systems in cells at a physiological level, neutralize free radical reactions, chelate iron ions and inhibit lipid peroxidation. The chelation of iron ions by phenolic compounds is particularly important because iron can catalyze the formation of ROS through the Fenton reaction. Inhibiting this reaction can help prevent the formation of harmful ROS and reduce oxidative stress in cells [9,43,44,45,46,47].

The phenolic groups are radical scavengers and form highly stable phenoxy radicals, thereby interrupting oxidation chain reactions in the cell [48]. The therapeutic effects of phenolic compounds, which are attributed to their antioxidant properties, have important pharmacological implications in the treatment of a wide range of diseases characterized by changes in cell metabolism and growth. However, the oxidative stress-modulating effects of phenolic compounds are not limited to their antioxidant properties, as observed in normal tissues with the physiological pH. It is well known that this parameter is shifted to the acidic side in the microenvironment of tumors, which is the main cause of the migration and invasion of neoplastic cells [49,50]. Under such conditions, phenols begin to exhibit pro-oxidant properties, which induce the apoptotic death of transformed cells. This is due to their auto-oxidation in such an environment, resulting in the formation of a complex combination of semiquinones and quinones. Additionally, considering the fact that tumor cells exhibit reprogramming of transition metal metabolism and, as a result, excessively high levels of iron or copper ions, the oxidation of phenols under such conditions also leads to the formation of radicals, including highly reactive ones such as superoxide anion radicals [51,52]. Such pro-oxidant activity of phenolic compounds can induce lipid peroxidation, DNA damage and cell apoptosis. It should be emphasized that the pro-oxidant properties of phenolic compounds, which lead to aberrant generation of ROS to a cytotoxic level, are observed specifically in tumor cells but not in normal cells due to differences in transition metal concentrations and metabolic activity [47]. Thus, the switching of the antioxidant properties of phenolic compounds observed in a healthy microenvironment to pro-oxidant action is an undeniable advantage in developing therapeutic strategies for the treatment of malignant neoplasms, providing selectivity for antitumor effects and avoiding side effects.

It is precisely due to the above-described properties of phenolic compounds that the results obtained for phosphorylated SHP hybrids in our study can be explained. Analysis of the effect of the synthesized compounds on the process of lipid peroxidation in the rat brain homogenates, which essentially represents a model reproducing normal conditions for the action of phenols, showed the presence of antioxidant activity for all investigated substances. In turn, analysis of the extracellular content of ROS directly in HuTu tumor cells using flow cytometry showed a significant increase in free radicals under the action of **11b** and **17b**. The obtained data confidently confirm the hypothesis of the paradoxical dual action of phenolic compounds on processes related to oxidative stress depending on the conditions.

As one of the mechanisms of cytotoxic action detected for phosphorylated SHPs containing urea aryl and heterocyclic fragments, the ability of **11b** and **17b** to inhibit the process of glycolysis was revealed. Such an ability to modulate the metabolism of tumor cells is likely to be associated with the inclusion of urea/thiourea fragments in their structure. A data series indicates disruptions in glucose homeostasis caused by direct exposure to urea, which is associated with a decrease in its utilization due to inhibition of the activity of key glycolysis enzymes [47]. Furthermore, derivatives of urea such as N,N′-bis(3,5-dichlorophenyl)urea have been shown to induce apoptotic cell death in several lines of tumor cells by modulating mitochondrial function. This includes uncoupling mitochondrial oxidative phosphorylation and depolarizing the mitochondrial membrane, which activates the transport of various molecules through the lipid bilayer and leads to the leakage of pro-apoptotic factors from the internal contents of organelles [53]. Thus, inclusion of a pharmacophore fragment linked to urea in a complex molecule may be the reason for the formation of a multi-target agent with high antitumor potential.

The conducted electrochemical experiments have also established that the biological activity of SHPs is related to the overall redox capacity of the molecule. It is important to note that oxidation at the first potential of the OH group provides the best characteristics for antioxidant and antitumor activity, as other important fragments that bind to DNA molecules are not affected by oxidation. For the studied compounds that showed good selectivity and high cytotoxicity, their wave character in semi-differential cyclic voltammetry was significantly unchanged when pH was reduced.

The structure–activity relationship (SAR) of a series of new diarylmethylphosphonates combining SHP and thiourea/urea moieties (Figure 12) showed that derivatives based on thiourea, unlike urea, do not exhibit cytotoxicity towards M-Hela, MCF-7 and HuTu80 tumor cells, with the exception of compound **18c**. Interestingly, the tested aryl-substituted urea derivatives exhibit better cytotoxicity than phenyl ureas. Compounds **11b** and **17b**, containing Cl^−^ and NO_2_ groups in the para position of phenylurea, showed lower cytotoxicity towards the Chang liver normal cell line while maintaining high cytotoxicity towards the HuTu 80 tumor line with a selectivity index (SI) above three. In addition, the same **11b** and **17b** compounds, containing a nitro group and a chlorine atom, also showed the best ability to inhibit the process of glycolysis. If we evaluate the influence of substituents on the phosphorus atom in terms of the activity of the obtained series of diarylmethylphosphonates, it can be noted that the presence of a diethylphosphoryl fragment is beneficial.

Our studies showed the promise of compounds exhibiting antitumor properties on the basis of new diarylmethyl phosphonates combining SHP and urea fragments. The lead compounds in the series of **11b** and **17b** have a confirmed mechanism of cytotoxic action and show great potential for further optimization.

## 4. Materials and Methods

### 4.1. Materials

#### Chemicals

The reagents and solvents used for all activities presented in this research were purchased from local suppliers. Dialkyl/diphenyl[(3,5-di-*tert*-butyl-4-hydroxyphenyl)(2,4-diaminophenyl)methyl]phosphonate **2a**,**b**,**d** (1.0 mmol) or dialkyl/diphenyl[(3,5-di-tert-butyl-4-hydroxyphenyl)(2,6-diaminopyridin-3-yl)methyl]phosphonate **3a**–**d** were synthesized according to the literature [37,54].

### 4.2. General Procedures for Compound Identification

The ^1^H- and ^13^C-NMR spectra were recorded on a Bruker AVANCE 400 spectrometer (Bruker BioSpin, Rheinstetten, Germany) operating at 400 MHz (for ^1^H NMR), 101 MHz (for ^13^C NMR) and 162 MHz (for ^31^P NMR); Brucker AVANCEIII-500 spectrometers (Bruker Corporation, Rheinstetten, Germany) operating at 500 MHz (for ^1^H NMR) and 126 MHz (for ^13^C NMR); and Brucker AVANCEIII-600 spectrometers (Bruker Corporation, Rheinstetten, Germany) operating at 600.13 MHz (for ^1^H NMR), 150.19 MHz (for ^13^C NMR) and 242.94 MHz (for ^31^P NMR). Chemical shifts were measured in δ (ppm) with reference to the solvent (δ = 2.56 ppm and 39.52 ppm for DMSO-d_6_ for ^1^H and ^13^C NMR, respectively). IR spectra were recorded on an IR Fourier spectrometer Tensor 37 (Bruker Optik GmbH, Germany) in the 400–3600 cm^−1^ range in KBr. The mass spectra were obtained on a Bruker Daltonics MALDI TOF/TOF instrument. Elemental analysis was performed on a CHNS-O Elemental Analyser EuroEA3028-HT-OM (EuroVector S.p.A., Milan, Italy). The melting points were determined on a JK-MAM-4 Melting-point Apparatus with Microscope (SGW-X4 JINGKE SCIENTIFIC INSTRUMENT Co., Shanghai, China). The progress of reactions and the purity of products were monitored by TLC on Sorbfil UV-254 plates (Sorbpolimer, Krasnodar, Russia); the chromatograms were developed under UV light.

To a 5 mL chloroform solution of dialkyl/diphenyl[(3,5-di-*tert*-butyl-4-hydroxyphenyl)(2,4-diaminophenyl)methyl]phosphonate **2a**,**b**,**d** (1.0 mmol) or dialkyl/diphenyl[(3,5-di-*tert*-butyl-4-hydroxyphenyl)(2,6-diaminopyridin-3-yl)methyl]phosphonate **3a**–**d** (1.0 mmol), the phenyl isocyanate (8.0 mmol), respective aryl isocyanate (8.0–12.0 mmol) or phenyl isothiocyanate (8.0 mmol) was added. The resulting solutions were stirred on a magnetic plate until a precipitation formed at ambient temperature for 4 h (**5a**–**d**; **6a**,**b**,**d**), 12 h (**8a**–**d**) or 3 h (**9a**–**d**; **10a**–**d**; **11c**,**d**; **12c**,**d**; **13a**,**b**,**d**–**17a**,**b**,**d**), while **18a**–**d** and **19a**,**b**,**d** were heated under reflux for 15 h. The precipitate was filtered off, washed once with ethyl acetate and hexane and dried under vacuum (0.06 mm Hg) at 40 °C to constant weight. Synthetic procedures, compound characterization data (Appendix A) and the ^1^H-,^31^P-,^13^C-NMR spectra of compounds **5a**–**d**; **6a**,**b**,**d**; **7a**,**b**; **8a**–**d**–**10a**–**d**; **11c**,**d**; **12c**,**d**; **13a**,**b**,**d**–**17a**,**b**,**d**; **18a**–**d** and **19a**,**b**,**d** are included in the Appendix A.

### 4.3. Biology

#### 4.3.1. Cells and Materials

For the experiments, we used tumor cell cultures M-HeLa clone 11 (epithelioid carcinoma of the cervix, subline HeLa., clone M-HeLa), HuTu 80 (human duodenal adenocarcinoma) and MCF7 (human breast adenocarcinoma (pleural fluid)) from the collection of the Institute of Cytology, Russian Academy of Sciences (St. Petersburg); human liver cells (Chang liver) from the collection of the Research Institute of Virology at the Russian Academy of Medical Sciences (Moscow); and SH-SY5Y (neuroblastoma)from the Laboratory of Tumor Cell Genetics at the Scientific Research Institute of Carcinogenesis N.N. Blokhin National Medical Research Center of Oncology.

#### 4.3.2. Cytotoxicity Assay

The cytotoxic effect on cells was determined using the colorimetric method of cell proliferation—the MTT test. NADP-H-dependent cellular oxidoreductase enzymes can, under certain conditions, reflect the number of viable cells. These enzymes are able to reduce the tetrazolium dye (MTT)-3-(4,5-dimethylthiazol-2-yl)-2,5-diphenyl-tetrazolium bromide to insoluble blue-violet formazan, which crystallizes inside the cell. The amount of formazan formed is proportional to the number of cells with active metabolism [55].

Cells were seeded on a 96-well Nunc plate at a concentration of 5 × 10^3^ cells per well in a volume of 100 μL of medium and cultured in a CO_2_ incubator at 37 °C until a monolayer was formed. The nutrient medium was then removed and 100 µL solutions of the test drug at the given dilutions were added to the wells, which were prepared directly in the nutrient medium with the addition of 5% DMSO to improve solubility. After 48 h, the nutrient medium was removed from the plates and 100 µL of the nutrient medium without serum with MTT at a concentration of 0.5 mg/mL was added and incubated for 4 h at 37 °C. Formazan crystals were added to 100 µL of DMSO in each well. Optical density was recorded at 540 nm on an in vitro logic microplate reader (Russia). The experiments for all compounds were repeated in triplicate.

#### 4.3.3. Flow Cytometry Assay

Cell Culture. M-HeLa cells at 1 × 10^6^ cells/well in a final volume of 2 mL were seeded into six-well plates. After 48 h of incubation, various concentrations of test compounds were added to wells.

Cell Apoptosis Analysis. The cells were harvested at 2000 rpm for 5 min and then washed twice with ice-cold PBS, followed by resuspension in binding buffer. Next, the samples were incubated with 5 μL of annexin V- Alexa Fluor 647 (Sigma-Aldrich, St. Louis, MO, USA) and 5 μL of propidium iodide for 15 min at room temperature in the dark. Finally, the cells were analyzed by flow cytometry (Guava easy Cyte, Merck, Rahway, NJ, USA) within 1 h. The experiments were repeated in triplicate.

Mitochondrial Membrane Potential. Cells were harvested at 2000 rpm for 5 min and then washed twice with ice-cold PBS, followed by resuspension in JC-10 (10 µg/mL) and incubation at 37 °C for 10 min. After the cells were rinsed three times and suspended in PBS, the JC-10 fluorescence was observed by flow cytometry (Guava easy Cyte, Merck, USA).

Detection of Intracellular ROS. M-HeLa cells were incubated with compound **17b** at concentrations of IC_50/2_ and IC_50_ for 48 h. ROS generation was investigated using flow cytometry assays and the CellROX^®^ Deep Red flow cytometry kit. For this, M-HeLa cells were harvested at 2000 rpm for 5 min and then washed twice with ice-cold PBS, followed by resuspension in 0.1 mL of medium without FBS, to which was added 0.2 μL of CellROX^®^ Deep Red before incubation at 37 °C for 30 min After washing the cells three times and suspending them in PBS, the production of ROS in the cells was immediately monitored using a flow cytometer (Guava easy Cyte, Merck, USA).

#### 4.3.4. Assessment of Lipid Peroxidation (LPO)

Rat brain homogenates. Whole brain homogenates (≈2.5 mg protein/mL) were obtained from 3-month-old male rats. Rats were decapitated under CO_2_ anesthetic conditions and the whole brain tissue was quickly isolated and placed on ice. Brain samples were washed and homogenized in buffer containing KCl (120 mM) and HEPES (20 mM) at pH = 7.4 using a Teflon glass homogenizer. The obtained material was then centrifuged at 4000 rpm for 10 min and a supernatant was collected, which was stored at a temperature of 4 °C until further lipid peroxidation analysis.

TBARS test. The lipid peroxidation assay was carried out using the modified method as described earlier [56]. The optical density of the selected supernatant containing malonic dialdehyde was measured on a plate analyzer (Cytation3, BioTech Instruments Inc., Winooski, VT, USA) at λ = 540 nm.

#### 4.3.5. Membrane Potential of Isolated Mitochondria

The membrane potential of rat brain mitochondria was measured with a Victor 3 plate analyzer (Perkin Elmer, Hamburg, Germany) using the potential-dependent indicator Safranin A [57]. The mitochondrial preparation was diluted in a buffer containing mannitol (225 mM) (Dia-M, Moscow, Russia), sucrose (75 mM) (Sigma Aldrich, Saint Louis, MO, USA), HEPES (10 mM) (Gibco, Scotland, UK), EGTA (20 µM) (Dia-M, Moscow, Russia) and KH_2_PO_4_ (1 mM) at pH = 7.4 at a ratio of 0.5 mg of protein to 1 mL of medium. Safranin A (5 µM) was added to the suspension immediately before the start of the measurement. The organelles were then energized by potassium succinate (5 mM) in the presence of rotenone (1 µM). Mitochondrial pore opening was induced by the addition of CaCl_2_ (25 µM).

#### 4.3.6. Glycolysis Flux Assay

The ability of synthesized compounds to modulate anaerobic glycolysis was studied using the Agilent Seahorse XF96e Analyzer (Seahorse Bioscience, Billerica, MA, USA) on the neuroblastoma SH-SY5Y cell line (40,000 cells per well) as described earlier [56].

Glycolysis was measured by subtracting the maximum rate of extracellular acidification of the medium before glucose injection from the maximum rate before oligomycin injection. Glycolytic capacity was measured by subtracting the maximum velocity value before glucose injection from the maximum velocity value after oligomycin injection. Glycolytic reserve was defined as the difference between glycolytic capacity and glycolysis.

#### 4.3.7. Statistical Analysis

IC_50_ values were estimated using the Quest Graph IC_50_ Calculator (AAT Bioquest, Inc., Sunnyvale, CA, USA) (Version 2022) (accessed on 25 June 2022) [58].

### 4.4. Electrochemical Measurements

Electrochemical measurements were taken on a BASi Epsilon EClipse electrochemical analyzer (West Lafayette, IN, USA). The program concentrated cEpsilon-ECUSB-V200 waves at the potential scan rate t = 100 mV∙s^−10^ in a CH_3_CN 0.1 M solution of Bu_4_NBF_4_ at 295 K. A glassy carbon working electrode (ð = 3 mm) embedded in Teflon and a Pt wire as the counter electrode were used in the electrochemical cell. Before each measurement, the surface of the working electrode was mechanically polished. Ag/AgCl (0.01 M KCl) was used as a reference electrode. The reference electrode was connected with the cell solution by a modified Luggin capillary filled with the supporting electrolyte solution (0.1 M Bu_4_NBF_4_ in CH_3_CN). Thus, the reference electrode assembly had two compartments, each terminated with an ultra-fine glass frit to separate the AgCl from the analyte. The scan rate was 100 mV∙s^−1^. The measurements were performed in a temperature-controlled electrochemical cell (volume from 5 mL to 20 mL) in an inert gas atmosphere (N_2_).

### 4.5. Molecular Docking

The corresponding structures were found in the PDB protein structure bank for pyruvate kinase and hexokinase—4G1N and 5HEX. The structures were loaded and processed with the Protein Preparation Wizard subprogram of the Schrodinger Suite [59,60,61,62,63,64,65,66,67,68,69,70,71]. Missing loops and side chains, if any, were restored; preprocessing was performed using the Prime module [60,61,62,63,64], with hydrogen bonds optimized; non-key water and other non-key small molecules, if any, were removed; and limited minimization of protein geometry was performed using the OPLS3e force field [65]. 

Ligand structures were prepared using the subprogram LigPrep [66].

Docking to the active centers of proteins was performed using the Induced Fit Docking protocol [64,67,68,69,70,71], and redocking of the reference ligand was performed to verify the chosen method. Active centers for Induced Fit Docking were declared as cubes centered at the geometric coordinate center of the corresponding co-crystallized ligands. The edge of the cubes corresponded to the possibility of docking ligands co-crystallized with the reference protein. Amino acids within a 5 Å radius of the ligand atoms were processed by the Prime module based on the results of the pre-docking of ligands in the Induced Fit algorithm. Glide docking post-processing was performed using the standard precision protocol.

The RMSD of the reference ligand of the 5HEX structure was 0.7096, indicating that the reference pose is reproduced well and the docking method to the active center of hexokinase protein is chosen well.

The RMSD of the reference ligand of the 4G1N structure was 1.0080, indicating that the reference pose is reproduced satisfactorily and that the method of docking into the active center of the pyruvate kinase protein is selected satisfactorily enough.

## 5. Conclusions

A series of novel anticancer smart agents were created by attaching a phenol (SHP), a diaryl urea and a phosphonate group at the same carbon. This combination of complementary pharmacophores resulted in interesting and useful properties for the resulting family of molecules. Several of the 45 new hybrid compounds of this family show high cytotoxic activity against tumor cells. In the study of the mechanisms of toxic action, it was found that substances that reduce the survival of neoplastic cells are capable of inhibiting the process of glycolysis in SH-SY5Y neuroblastoma cells by blocking the allosteric center of pyruvate kinase M2, thus leading to powerful dissipation of the mitochondrial membrane and causing an energy collapse.

It is interesting to note that the discovered selective toxic effect for some compounds, specifically against the tumor microenvironment, may be due to the presence of a sterically hindered phenol fragment in the molecule, which allows for switching of the antioxidant properties exhibited by substances under normal conditions to a pro-oxidant effect in HuTu 80 tumor cells through the significantly increased production of free radicals.

In addition, during electrochemical experiments, it was established that such an effect is associated not only with the oxidative potential of the OH group, but also with the overall redox capacity of the entire molecule. For compounds with the most promising profile of antitumor activity identified as a result of the “structure–activity” analysis (**11b** and **17b**), an additional ability to induce apoptosis in neoplastic cells through the intrinsic mitochondrial pathway was discovered.

Thus, the lead compounds **11b** and **17b** can be considered as promising substances for further chemical optimization aimed at creating advanced antitumor smart agents based on them with improved characteristics.

## Data Availability

Data is contained within the article and Appendix A.

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
