# Peer review of "Hybrids of Sterically Hindered Phenols and Diaryl Ureas: Synthesis, Switch from Antioxidant Activity to ROS Generation and Induction of Apoptosis"

_ijms, 2023, doi:10.3390/ijms241612637_

Round 1
Reviewer 1 Report
In this paper the authors have shown design and synthesis of a series of novel anticancer smart agents were created by attaching a phenol (SHP), a diaryl urea and a phosphonate group.
Detailed studies were performed to validate the claims.
This paper can be accepted in its current form.
Author Response
Response to Reviewer 1 Comments
Q1 In this paper the authors have shown design and synthesis of a series of novel anticancer smart agents were created by attaching a phenol (SHP), a diaryl urea and a phosphonate group.
Detailed studies were performed to validate the claims.
This paper can be accepted in its current form.
Reply: Thank you very much for such positive comments. We are pleased to hear that the paper on the design and synthesis of novel anticancer smart agents has been thoroughly studied and can be accepted in its current form. We appreciate your time and effort in reviewing the manuscript.
Reviewer 2 Report
In this manuscript by Gibadullina et al. described the development of antitumor compounds which have biological activity (cytotoxic effects to tumor cells) but only exhibit antioxidant effects to normal cells. The investigation of these synthetic compounds has been comprehensively studied and very insightful, and suitable for publication. There are only a few questions about the manuscript.
1. The authors are suggested to provide experimental details in the electrochemical experiments such as potential range, scan rate, and reference electrode used.
2. Line 551, SPH/urea hybrid should be SHP/urea hybrid.
3. In Figure S3, when 30eq acid is added to compound 13 b, it is also seen the reduction peak observed approximately at 0.6 V, while it is not observed when compound 14b added with 30 eq. What is the explanation for this?
4. In Figure S6, when compound 17b was added with the increasing acid until 30eq, we can see the intensity of oxidation peak approximately at 2.25 V gradually diminished. What is the explanation for this?
5. In Figure 11, there are 2 reduction peaks observed when compound 13b was scanned from 2.5 to 0 V. Are these 2 peaks both contributing to the reduction of C=O to -OH? And why does the intensity of these reduction peaks different from each other?
The studies performed in this work have been done comprehensively and well-structured. I have learned something new from this work and this is very insightful for me. Thank you for your trust to review this manuscript.
Author Response
Response to Reviewer 2 Comments
In this manuscript by Gibadullina et al. described the development of antitumor compounds which have biological activity (cytotoxic effects to tumor cells) but only exhibit antioxidant effects to normal cells. The investigation of these synthetic compounds has been comprehensively studied and very insightful, and suitable for publication. There are only a few questions about the manuscript.
Q1 The authors are suggested to provide experimental details in the electrochemical experiments such as potential range, scan rate, and reference electrode used.
Reply: Thank you for your comment. The description has been updated.
.
Q2 Line 551, SPH/urea hybrid should be SHP/urea hybrid.
Reply: Thank you for finding the typo. Corrected.
Q3 In Figure S3, when 30eq acid is added to compound 13 b, it is also seen the reduction peak observed approximately at 0.6 V, while it is not observed when compound 14b added with 30 eq. What is the explanation for this?
Reply: Most likely, the origin of the difference in the electrochemical behavior of compounds 13b and 14b is complex and may not originate from a single factor. However, we suggest that the location of the Me group in the meta position of compound 13b may play a role in its reversible oxidation behavior. Further research may be needed to determine the exact cause of this difference.
Q4 In Figure S6, when compound 17b was added with the increasing acid until 30eq, we can see the intensity of oxidation peak approximately at 2.25 V gradually diminished. What is the explanation for this?
Reply: In the case of compound 17b in Figure S6, the intensity of the oxidation peak at 2.25 V gradually diminished as the acid concentration increased up to 30eq. We think that it indicates that this compound was being partially oxidized in the presence of the acid.
Q5 In Figure 11, there are 2 reduction peaks observed when compound 13b was scanned from 2.5 to 0 V. Are these 2 peaks both contributing to the reduction of C=O to -OH? And why does the intensity of these reduction peaks different from each other?
Reply: It is possible that both of these peaks are contributing to the reduction of C=O to -OH. Regarding the difference in intensity between the two peaks, it is possible that this is due to a difference in the accessibility of the two reduction sites. For example, one site may be more sterically hindered or less electron-rich than the other, which could affect the rate of reduction and the intensity of the corresponding peak.
The studies performed in this work have been done comprehensively and well-structured. I have learned something new from this work and this is very insightful for me. Thank you for your trust to review this manuscript.
Reply: Thank you for your comment. We appreciate your time and effort in reviewing the manuscript. We are very pleased to hear the kind words about this work, especially since we have put a lot of time and effort into this work.
Reviewer 3 Report
The manuscript describes the synthesis and biological evaluation of a class of hindered pheonls. The relatively large number of authors allows the paper to be executed very well in all aspects of the study including synthesis, chemical characterization, electrochemical behavior, quantum chemical calculations, cytotoxicity measurements, and biological mechanism-of-action experiments. The study would be an excellent addition to IJMS.
Analysis:
1. The introduction provides a compelling story for the molecules chosen to study.
2. The chemical synthesis and characterization, including supporting information, is expertly conducted. One comment is that typically 13C chemical shifts are reported to one digit after the decimal point, unless an additional
digit will help distinguish overlapping peaks.
3. The cytotoxic profiles are well-described with appropriate replicates, error reporting, and controls.
4. The in vitro and intracellular ROS assays are appropriate experiments to complete the biological assessment of these compounds based on their assumed redox active functionality. The claims in the paper are supported by the data.
5. The discussion is logical and well-written.
Recommendations:
6. The mechanism of action of sterically hindered phenols shown in Figure 1E and the accompanying text (lines 99-109) are in contradiction to the generally accepted mechanism of BHT-ROS mode-of-action. Interestingly, this is acknowledged in lines 94 and 95 "... significantly affect the rate of initial hydrogen atom transfer." Therefore, the importance of the benzyllic deprotonation prior to hydrogen abstraction is confusing. Correspondingly, the pKa of the benzyllic proton is still likely above that of the phenol (pka of BHT =12). This discussion should be altered to acknowledge the typical sequence of events for phenol-ROS behavior with this line of discussion maybe being presented as a hypothesis.
7. Computational docking of the SHP was conducted with pyruvate kinase and hexokinase for the most active compounds (lines 439-440). However, why these two enzymes were chosen is not explained. It is left to the reader to deduce that it is because they are important phosphryl transfer enzymes in glycosylsis or potentially for a more validated reason.
8. Structures in figure 10 are inconsistent in format to the other structures in the paper. They also are at a particular poor resolution. These structures should be improved.
Typos
line 3: Ros -> ROS
Line 216: Merk -> Merck
Line 321: 3.2.1. Сytotoxic Profile
The manuscript is well-written with logical arguments and few typographical errors.
Author Response
Response to Reviewer 3 Comments
The manuscript describes the synthesis and biological evaluation of a class of hindered pheonls. The relatively large number of authors allows the paper to be executed very well in all aspects of the study including synthesis, chemical characterization, electrochemical behavior, quantum chemical calculations, cytotoxicity measurements, and biological mechanism-of-action experiments. The study would be an excellent addition to IJMS.
Analysis:
- The introduction provides a compelling story for the molecules chosen to study.
- The chemical synthesis and characterization, including supporting information, is expertly conducted. One comment is that typically 13C chemical shifts are reported to one digit after the decimal point, unless an additional digit will help distinguish overlapping peaks.
- The cytotoxic profiles are well-described with appropriate replicates, error reporting, and controls.
- The in vitro and intracellular ROS assays are appropriate experiments to complete the biological assessment of these compounds based on their assumed redox active functionality. The claims in the paper are supported by the data.
- The discussion is logical and well-written.
Recommendations:
Q1 6. The mechanism of action of sterically hindered phenols shown in Figure 1E and the accompanying text (lines 99-109) are in contradiction to the generally accepted mechanism of BHT-ROS mode-of-action. Interestingly, this is acknowledged in lines 94 and 95 "... significantly affect the rate of initial hydrogen atom transfer." Therefore, the importance of the benzyllic deprotonation prior to hydrogen abstraction is confusing. Correspondingly, the pKa of the benzyllic proton is still likely above that of the phenol (pka of BHT =12). This discussion should be altered to acknowledge the typical sequence of events for phenol-ROS behavior with this line of discussion maybe being presented as a hypothesis.
Reply: Thank you for your comment. This is a very important point and we appreciate an opportunity to discuss it further. The story has two parts. In the first of them, as the reviewer has summarized very well, is the generally accepted mechanism of BHT-ROS interaction that involves hydrogen transfer from the phenol OH to the an external radical. This reaction breaks radical chains and accounts for the antioxidative properties of phenols. However, there is the 2nd part of the story, i.e. behavior of phenols under the increased oxidative stress conditions. Our data clearly show that there these molecules behave as oxidants. For that, they have to transform into quinone methides or their radical-anions, the process that, unavoidably, breaks the benzylic C-H as well. In reality, it may not be very important if the benzylic C-H is activated before or after the OH. In any case, there is no doubt that the C-H activation can proceed much more readily after HAT from the OH group because O-radical is powerful acceptor that greatly acidifies the para CH2Ar moiety. Hence, not only deprotonation makes the OH weaker but HAT from OH changes the C-H acidity as well.
Q2 7. Computational docking of the SHP was conducted with pyruvate kinase and hexokinase for the most active compounds (lines 439-440). However, why these two enzymes were chosen is not explained. It is left to the reader to deduce that it is because they are important phosphryl transfer enzymes in glycosylsis or potentially for a more validated reason.
Reply: Thank you for your comment. We expanded the rationale for the selection of the present enzymes for the docking procedure in the text of the manuscript and included several present-day references to supporting our choice.
Q3 8. Structures in figure 10 are inconsistent in format to the other structures in the paper. They also are at a particular poor resolution. These structures should be improved.
Reply: Corrected.
Q4 Typos
line 3: Ros -> ROS
Line 216: Merk -> Merck
Line 321: 3.2.1. Сytotoxic Profile
Reply: Thank you very much for your comments. Corrected. We will certainly consider your comments when preparing future manuscripts. We appreciate your time and effort in reviewing the manuscript.
Reviewer 4 Report
In the present study the authors tried to establish the SAR of a series of new diarylmethylphosphonates containing sterically hindered phenols (SHP) and thiourea/urea moieties for cytotoxic activity in vitro against tumor lines. Their studies showed the promise of compounds exhibiting antitumor properties on the basis of these new fragments. The lead compounds 11b and 17b have a confirmed mechanism of cytotoxic action and show great potential for further optimization. The authors successfully utilized SHPs in the drug design based on their chameleonic ability to switch from an antioxidant that can protect healthy tissues to highly cytotoxic species that can target tumor cells. The designing of these molecules was well planned so that the acidic benzylic C-H bond activation being a key aspect using the corresponding substituents.
The authors are successful in unravelling the promiscuity of the compounds exhibiting antitumor properties on the basis of new diarylmethyl phosphonates combining SHP and urea fragments. The mechanism of cytotoxic action with great potential for further optimization confirmed the paradoxical nature of these substances which combine pronounced antioxidant properties in radical trapping assays with increased reactive oxygen species generation in tumor cells.
Molecular docking studies, biological evolution and cytotoxic profile showed very reliable and promising. The compounds are not only well characterized but also kept an amazing effort on assigning the protons and carbons by using 2D NMR. Also, the authors tried to provide maximum possible insights. Overall, the manuscript can be accepted and publishable in IJMS provided after eliminating few typo errors.
Author Response
Response to Reviewer 4 Comments
In the present study the authors tried to establish the SAR of a series of new diarylmethylphosphonates containing sterically hindered phenols (SHP) and thiourea/urea moieties for cytotoxic activity in vitro against tumor lines. Their studies showed the promise of compounds exhibiting antitumor properties on the basis of these new fragments. The lead compounds 11b and 17b have a confirmed mechanism of cytotoxic action and show great potential for further optimization. The authors successfully utilized SHPs in the drug design based on their chameleonic ability to switch from an antioxidant that can protect healthy tissues to highly cytotoxic species that can target tumor cells. The designing of these molecules was well planned so that the acidic benzylic C-H bond activation being a key aspect using the corresponding substituents.
The authors are successful in unravelling the promiscuity of the compounds exhibiting antitumor properties on the basis of new diarylmethyl phosphonates combining SHP and urea fragments. The mechanism of cytotoxic action with great potential for further optimization confirmed the paradoxical nature of these substances which combine pronounced antioxidant properties in radical trapping assays with increased reactive oxygen species generation in tumor cells.
Molecular docking studies, biological evolution and cytotoxic profile showed very reliable and promising. The compounds are not only well characterized but also kept an amazing effort on assigning the protons and carbons by using 2D NMR. Also, the authors tried to provide maximum possible insights. Overall, the manuscript can be accepted and publishable in IJMS provided after eliminating few typo errors.
Reply: Thank you very much for your comments. We appreciate your time and effort in reviewing the manuscript.